# SUPPRESSING OUTLIER RECONSTRUCTION IN AUTOENCODERS FOR OUT-OF-DISTRIBUTION DETECTION

## ABSTRACT

While only trained to reconstruct training data, autoencoders may produce high-quality reconstructions of inputs that are well outside the training data distribution. This phenomenon, which we refer to as outlier reconstruction, has a detrimental effect on the use of autoencoders for outlier detection, as an autoencoder will misclassify a clear outlier as being in-distribution. In this paper, we introduce the Energy-Based Autoencoder (EBAE), an autoencoder that is considerably less susceptible to outlier reconstruction. The core idea of EBAE is to treat the reconstruction error as an energy function of a normalized density and to strictly enforce the normalization constraint. We show that the reconstruction of non-training inputs can be suppressed, and the reconstruction error made highly discriminative to outliers, by enforcing this constraint. We empirically show that EBAE significantly outperforms both existing autoencoders and other generative models for several out-of-distribution detection tasks.

## 1 INTRODUCTION

An autoencoder (Rumelhart et al., 1986) is a neural network trained to reconstruct samples from a training data distribution. As the quality of reconstruction is expected to degrade for inputs that are significantly different from training data, autoencoders are widely used in outlier detection (Japkowicz et al., 1995) where an input with a large reconstruction error is classified as out-of-distribution (OOD). Such autoencoders for outlier detection have been applied in domains ranging from video surveillance (Zhao et al., 2017) to medical diagnosis (Lu & Xu, 2018).

Contrary to widely-held belief, autoencoders are in fact capable of accurately reconstructing outliers, casting doubt on their reliability as an outlier detector. Lyudchik (2016) showed that an autoencoder trained on MNIST with the digit seven excluded can reconstruct an image of the excluded digit, and Tong et al. (2019) reported that an autoencoder trained on MNIST can reconstruct an image with all zero pixels. The reconstruction of outliers is also observed for non-image data (Zong et al., 2018).

In this paper, we investigate this unexpected behavior of autoencoders more deeply, which we refer to as *outlier reconstruction*. In the course of our investigation, we reproduce the findings of Lyudchik (2016) and Tong et al. (2019), and additionally discover other interesting cases (Figure 1). Our experiments suggest that outlier reconstruction is not a fortuitous artifact of stochastic training but is, in fact, a consequence of inductive biases inherent in an autoencoder.

Outlier reconstruction should be suppressed for an autoencoder-based outlier detector, since a reconstructed outlier undermines the detector's performance by being mistaken to be an inlier. Despite the long history of autoencoder research (Rumelhart et al., 1986; Bank et al., 2020), the outlier reconstruction phenomenon has only recently begun to receive attention (Lyudchik, 2016; Tong et al., 2019; Zong et al., 2018), with few works explicitly proposing solutions to the outlier reconstruction problem (Gong et al., 2019). Previous works focused on regularization techniques that prevent an autoencoder from being an identity mapping (and thus reconstructing all inputs). However, outlier reconstruction still occurs in popular regularized autoencoders, including denoising autoencoders (DAE, Vincent et al. (2008)), variational autoencoders (VAE, Kingma & Welling (2014)), and Wasserstein autoencoders (WAE, Tolstikhin et al. (2017)), as we shall show in our experiments (Table 1).

Figure 1: (**Left**) Input images (first row), reconstructions from an autoencoder (AE, second row) and from EBAE (third row). AE and EBAE are trained on MNIST. (**Right**) Distribution of reconstruction errors for inliers (CIFAR10) and outliers (the other curves).

In this paper, we propose the **Energy-based Autoencoder (EBAE)**, an autoencoder in which the reconstruction of outliers is explicitly suppressed during training. In each training step of EBAE, "fake" samples with small reconstruction error are generated. These well-reconstructed fake samples serve as probes for potential reconstructed outliers. Then, EBAE *maximizes* the reconstruction errors of the generated samples, while *minimizing* the reconstruction errors of "real" training samples. When the generated samples become indistinguishable to training data, the gradients from the fake samples and real samples balance, and thus the training converges.

The training scheme naturally arises from defining a probability density for EBAE using its reconstruction error. The density of EBAE is given as $p_\theta(\mathbf{x}) = \exp(-E(\mathbf{x}))/\Omega$, where $E(\mathbf{x})$ is the reconstruction error of $\mathbf{x}$ and $\Omega$ is a normalization constant. This formulation of defining a density using a scalar function is often called an energy-based model in the literature (Mnih & Hinton, 2005; Hinton et al., 2006), and $E(\mathbf{x})$ is called the energy of the density. Maximizing likelihood in this formulation results in contrastive divergence learning (Hinton, 2002), which minimizes the energy of the training data while maximizing the energy of the samples from the model.

When generating samples with small reconstruction error during training, we use a novel sampling scheme specifically designed for EBAE. Our sampling scheme is based on Langevin Monte Carlo but leverages the latent space of an autoencoder to generate diverse samples which facilitates the training of EBAE.

Setting the reconstruction error as the energy, EBAE incorporates two major outlier detection criteria, large reconstruction error (Japkowicz et al., 1995) and low likelihood (Bishop, 1994), since the two are equivalent in EBAE. Generally, the two criteria do not necessarily overlap in other methods, e.g., VAE or energy-based models (Zhai et al., 2016). Recent studies show that a likelihood-based outlier detector using a deep generative model, such as an auto-regressive model or flow-based model, fails to correctly classify certain obvious outliers (Nalisnick et al., 2019; Hendrycks et al., 2019). However, EBAE is able to detect such outliers successfully while still using likelihood as the decision criterion.

The contributions of our paper can be summarized as follows:

- We report and investigate various cases of outlier reconstruction in autoencoders;
- We propose EBAE, an autoencoder significantly less prone to outlier reconstruction;
- We present a sampling method tailored for EBAE which efficiently generates diverse samples;
- We empirically show that EBAE is highly effective for outlier detection.

Section 2 provides a brief introduction on autoencoder-based outlier detection. In Section 3, we investigate outlier reconstruction in depth with illustrative examples. Section 4 describes EBAE. Related works are reviewed in Section 5. Section 6 presents experimental results. Section 7 concludes the paper.

## 2 BACKGROUND

### 2.1 PROBLEM SETTING

In this paper, we consider the outlier detection problem, which is also referred to as novelty detection, open set recognition, or OOD detection in literature. The goal is to classify outliers from

in-distribution samples, while no information regarding the outliers to be detected is provided during training. Formally, we are given a set of inliers $\mathbf{x} \in \mathcal{X} \subset \mathbb{R}^{D_{\mathbf{x}}}$ from the underlying data density function $p(\mathbf{x})$ for training. $D_{\mathbf{x}}$ is the dimensionality of $\mathbf{x}$ and $\mathcal{X}$ is the support of $p(\mathbf{x})$. An outlier is typically defined as a sample from the $\rho$-sublevel set of data density $\{\mathbf{x}|p(\mathbf{x}) \leq \rho\}$ (Steinwart et al., 2005). Note that a sample located outside of the support $\mathcal{X}$ belongs to the 0-sublevel set and hence is an outlier for all $\rho \geq 0$.

An outlier detection system typically produces a scalar decision function $c(\mathbf{x})$, and $\mathbf{x}$ is predicted as an outlier if $c(\mathbf{x}) > \eta$ for the threshold $\eta$. Setting the threshold controls the trade-off between false positive rate and false negative rate. In our experiments, we shall use area under receiver operating characteristic curve (AUC) as a threshold-independent performance metric when evaluating outlier detection algorithms.

## 2.2 AUTOENCODER-BASED OOD DETECTION

An autoencoder consists of an encoder $f_e(\mathbf{x}) : \mathbb{R}^{D_{\mathbf{x}}} \to \mathbb{R}^{D_{\mathbf{z}}}$ and a decoder $f_d(\mathbf{z}) : \mathbb{R}^{D_{\mathbf{z}}} \to \mathbb{R}^{D_{\mathbf{x}}}$, where $D_{\mathbf{z}}$ is the dimensionality of the latent vector $\mathbf{z}$. An input $\mathbf{x}$ is sequentially processed through the encoder and the decoder, producing its reconstruction $\tilde{\mathbf{x}} = f_d(f_e(\mathbf{x}))$. The reconstruction error is the discrepancy between $\mathbf{x}$ and $\tilde{\mathbf{x}}$. $L_2$ distance $||\mathbf{x} - \tilde{\mathbf{x}}||^2$ is a popular choice of discrepancy measure, but other error metrics are also applicable. A encoder and a decoder are deep neural networks and are jointly trained to minimize the mean reconstruction error of training data through stochastic gradient descent.

In an autoencoder-based outlier detection system (Japkowicz et al., 1995), the reconstruction error of an autoencoder trained on in-distribution samples is used as the decision function $c(\mathbf{x}) = ||\mathbf{x} - \tilde{\mathbf{x}}||^2$, and an input with large reconstruction error is classified as OOD. However, in the next section, we shall show that an autoencoder could unexpectedly produce very small reconstruction error for inputs not drawn from the training distribution. Thus an autoencoder-based outlier detection system may fail to detect such outliers.

## 3 OUTLIER RECONSTRUCTION

The outlier reconstruction is a phenomenon that an autoencoder unexpectedly succeeds in reconstructing an input even though it is located outside of the training distribution. In this section, we provide illustrative examples that show that outlier reconstruction is a consequence from the inductive biases of an autoencoder.

**Smoothness of mappings** When the training data distribution consists of multiple clusters, the outliers from the region between the clusters are likely to be reconstructed. Figure 2 depicts 2D synthetic data generated from a mixture of two disconnected uniform distributions and their reconstruction from autoencoders with one-dimensional latent space. The outliers (red crosses) from the middle of two clusters show reconstruction errors (the length of thin black lines) smaller than some inliers (blue dots). Tong et al. (2019) noted this type of outlier reconstruction and mentioned that outliers "close to the mean" of data or "in the convex hull" of data are likely to be reconstructed.

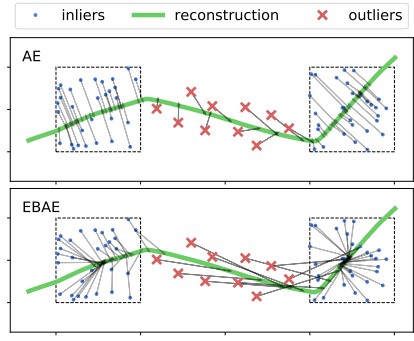

Figure 2: AE and EBAE trained on a bimodal distribution.

This phenomenon arises from the inductive bias of an autoencoder that its encoder and decoder are smooth mappings. The extreme case of this inductive bias can be found in linear principal component analysis (PCA). PCA, a linear counterpart of an autoencoder (Bourlard & Kamp, 1988), would reconstruct any outliers which reside on the principal axis. Note that this phenomenon is consistent with the objective function of an autoencoder and PCA, as the objective does not penalize the reconstruction of outliers.

**Compositionality** When there is a compositional structure in data, we can still observe a reconstructed outlier even if it lies outside of the convex hull of training data. The data are compositional

if each datum can be broken down into smaller reusable components; For example, MNIST can be considered highly compositional, since a digit image can be decomposed into smaller sub-patterns, such as straight lines and curves. An outlier can be successfully reconstructed when composed of a subset of components existing in the training data.

To demonstrate the effect of compositionality in outlier reconstruction, we make two synthetic datasets both of which are clearly out-of-distribution with respect to MNIST. The first dataset is HalfMNIST (the seventh column in Figure 1), consisting of MNIST images with a randomly chosen upper or lower half replaced by zero pixels. ChimeraMNIST (the last column in Figure 1), the second dataset, is a set of images generated by concatenating upper and lower halves of two randomly chosen digits. Although these images are not in the convex hull of MNIST digits, they share components found in MNIST. As shown in Figure 1, an autoencoder trained on MNIST have no problem reconstructing them. The classification AUCs from the reconstruction error are 0.482 for HalfMNIST and 0.69 for ChimeraMNIST, indicating poor classification.

It seems that an autoencoder learns to reconstruct each part of an image separately but is not able to judge whether the combination of the parts is valid as a whole. This compositional way of processing facilitates generalization of a model (Keysers et al., 2019), but the generalization of reconstruction in OOD inputs is not desirable for an autoencoder-based outlier detector.

**Distributed representation** We suspect the outlier reconstruction due to compositional processing may be attributed to the distributed representation (Mikolov et al., 2013) used in an autoencoder. To show the effect of the distributed representation, we train autoencoders on MNIST with the digit 9 excluded (MNIST-not9) and measure the reconstruction error of the digit 9 (MNIST9) under multiple values of latent dimensionality $D_{\mathbf{z}}$. Figure 3 shows the result. We observe the outlier reconstruction of

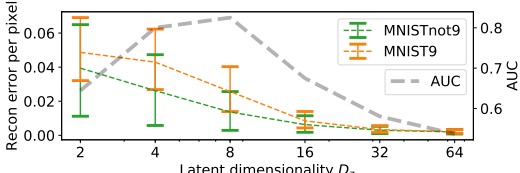

Figure 3: Reconstruction errors of MNISTnot9 and MNIST9. The error bars denote 80-percentile around the means.

MNIST9 possibly due to the compositional processing mentioned above. However, the outlier reconstruction occurs only when $D_{\mathbf{z}}$ is large. The latent representation is more distribution for large $D_{\mathbf{z}}$, as a larger number of hidden neurons are used to represent an input. This observation suggests that the distributed representation used in an autoencoder enables the compositional processing and thus facilitates outlier reconstruction.

Meanwhile, the model we propose shortly is not vulnerable to outlier reconstruction in all cases we examined above, as shown in Figure 1, Figure 2, and Table 1, even though it is based on the same autoencoder architecture.

## 4 ENERGY-BASED AUTOENCODERS

### 4.1 DEFINITION

We propose **Energy-based Autoencoder (EBAE)**, a generative model built from an autoencoder described in Section 2.2. The density of EBAE is designed to be large for an input with small reconstruction error. Formally, given an encoder $f_e(\mathbf{x})$ and a decoder $f_d(\mathbf{z})$, the density model of EBAE $p_\theta(\mathbf{x})$ is defined as follows:

$$p_\theta(\mathbf{x}) = \frac{1}{\Omega_\theta} \exp(-E_\theta(\mathbf{x})), \;\; E_\theta(\mathbf{x}) = ||\mathbf{x} - f_d(f_e(\mathbf{x}))||^2, \;\; \Omega_\theta = \int \exp(-E_\theta(\mathbf{x}))\mathrm{d}\mathbf{x} < \infty. \;\; (1)$$

To ensure the integral exists, we assume that the domain of $\mathbf{x}$ is bounded, and $\exp(-E_\theta(\mathbf{x}))$ is continuous and bounded. Often, the quantity $E_\theta(\mathbf{x})$ is referred to as an energy function, and the whole formulation is called energy-based model (Mnih & Hinton, 2005; Hinton et al., 2006).

As in a conventional autoencoders (Japkowicz et al., 1995), outliers are detected based on the large reconstruction error in EBAE. This decision is equivalent to determining an outlier based on low likelihood (Bishop, 1994), as the likelihood and the reconstruction error are in a linear relationship: $\log p_\theta(\mathbf{x}) = -E_\theta(\mathbf{x}) - \log \Omega_\theta$ where $\log \Omega_\theta$ is a constant with respect to $\mathbf{x}$.

## 4.2 TRAINING

EBAE is trained to maximize the likelihood of data. The expectation of the likelihood gradient with respect to the data density $p(\mathbf{x})$ is written as follows (the derivation in the supplementary material):

$$\mathbb{E}_{\mathbf{x} \sim p(\mathbf{x})}[\nabla_\theta \log p_\theta(\mathbf{x})] = -\mathbb{E}_{\mathbf{x} \sim p(\mathbf{x})}[\nabla_\theta E(\mathbf{x})] - \nabla_\theta \log \Omega_\theta \tag{2}$$

$$= -\mathbb{E}_{\mathbf{x} \sim p(\mathbf{x})}[\nabla_\theta E(\mathbf{x})] + \mathbb{E}_{\mathbf{x}' \sim p_\theta(\mathbf{x})}[\nabla_\theta E(\mathbf{x}')]. \tag{3}$$

The expectation $\mathbb{E}_{\mathbf{x} \sim p(\mathbf{x})}[\cdot]$ is with respect to training data, and the expectation $\mathbb{E}_{\mathbf{x}' \sim p_\theta(\mathbf{x}')}[\cdot]$ is with respect to samples generated from the model. Each gradient step would decrease the average reconstruction error of the training data while increase the average reconstruction error of "fake" data $\mathbf{x}'$. In practice, we approximate the expectations with a mini-batch of training data and generated samples in each iteration.

The second term $\mathbb{E}_{\mathbf{x}' \sim p_\theta(\mathbf{x})}[\nabla_\theta E(\mathbf{x}')]$ is responsible for inhibiting the reconstruction of outliers. The sampling process tend to generate high-likelihood samples which will produce small reconstruction error by definition. Therefore, the second term finds outliers which can be reconstructed by a current model and suppress them by applying the gradient. The training converges when $p_\theta(\mathbf{x})$ become identical to $p(\mathbf{x})$ as the two gradient terms are cancelled out. A converged EBAE will give small reconstruction errors only for in-distribution samples and large reconstruction errors for OOD inputs.

The suppression of outlier reconstruction in EBAE is originated from the enforcement of the normalization constraint, as the second gradient term is derived from the gradient of the normalization constant $\nabla_\theta \log \Omega_\theta$. Note that by removing the second term in Eq.(3), the expression reduces into the gradient of a conventional autoencoder, only minimizing the reconstruction error of inliers. Thus, from our energy-based formulation (Eq.(1)), a conventional autoencoder training does not properly maximizes the likelihood as it neglects normalization, and the lack of normalization is an explanation why outlier reconstruction occurs in autoencoders.

## 4.3 SAMPLING FROM EBAE

During training, samples are drawn from $p_\theta(\mathbf{x})$ by Markov Chain Monte Carlo (MCMC) method to approximate the second expectation in Eq.(3). For an energy function based on a deep neural network, Langevin Monte Carlo (LMC) (Welling & Teh, 2011; Neal et al., 2011) method is typically employed (Du & Mordatch, 2019; Grathwohl et al., 2020). However, we found that naive LMC mixes very slowly for the energy function of EBAE, possibly because the energy landscape is highly multi-modal.

Here, we propose a *two-stage* sampling scheme which is specifically designed for EBAE. The proposed method use the latent space structure in an autoencoder to traverse between energy basins. The scheme consists of two consecutive LMC chains, the **latent chain** and the **visible chain**, which are illustrated in Figure 5. The latent chain runs first in the latent space and aims to provide a good starting point for the visible chain, which runs in the visible space afterwards. After $T_\mathbf{z}$ steps of the latent chain, its sample $\mathbf{z}_{T_\mathbf{z}}$ is fed to the decoder, yielding the starting point for the visible chain $\mathbf{x}_0 = f_d(\mathbf{z}_{T_\mathbf{z}})$. The visible chain runs for $T_\mathbf{x}$ steps and produces the

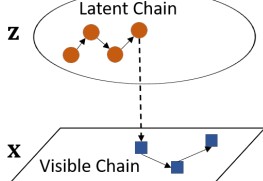

Figure 4: The illustration of two-stage sampling process.

sample. With the latent chain, the visible chain could start from diverse points with sufficiently high density, resulting in diverse samples with high quality.

The two chains can be formally written as follows:

$$\textbf{Latent chain: } \mathbf{z}_0 \sim \text{Uniform}, \ \mathbf{z}_{t+1} = \mathbf{z}_t + \frac{\lambda_\mathbf{z}}{2} \nabla_\mathbf{z} \log q_\theta(\mathbf{z}_t) + \epsilon_\mathbf{z}, \ \epsilon_\mathbf{z} \sim \mathcal{N}(\mathbf{0}_\mathbf{z}, \sigma_\mathbf{z}^2 \mathbf{I}_\mathbf{z}). \tag{4}$$

$$\textbf{Visible chain: } \mathbf{x}_0 = f_d(\mathbf{z}_{T_\mathbf{z}}), \ \mathbf{x}_{t+1} = \mathbf{x}_t + \frac{\lambda_\mathbf{x}}{2} \nabla_\mathbf{x} \log p_\theta(\mathbf{x}_t) + \epsilon_\mathbf{x}, \ \epsilon_\mathbf{x} \sim \mathcal{N}(\mathbf{0}_\mathbf{x}, \sigma_\mathbf{x}^2 \mathbf{I}_\mathbf{x}). \tag{5}$$

where $\mathbf{0}_\mathbf{z}, \mathbf{0}_\mathbf{x}, \mathbf{I}_\mathbf{z}$, and $\mathbf{I}_\mathbf{x}$ are zero vectors and identity matrices defined in the latent space or the input space, accordingly. The step sizes $\lambda_\mathbf{z}, \lambda_\mathbf{x}$ and the variances $\sigma_\mathbf{z}, \sigma_\mathbf{x}$ are tuned separately to achieve faster training in practice as done in Du & Mordatch (2019); Grathwohl et al. (2020). $p_\theta(\mathbf{z})$ is the density of EBAE defined in Eq.(1), and $q_\theta(\mathbf{z})$ is an *auxiliary density function* designed for the

latent chain to yield desirable starting point for the visible chain. We also define $q_\theta(\mathbf{z})$ through the energy-based formulation with an energy function $H_\theta(\mathbf{z})$:

$$q_\theta(\mathbf{z}) = \frac{1}{\Psi_\theta}\exp(-H_\theta(\mathbf{z})), \quad H_\theta(\mathbf{z}) = E_\theta(f_d(\mathbf{z})), \quad \Psi_\theta = \int \exp(-H_\theta(\mathbf{z}))\mathrm{d}\mathbf{z} < \infty. \quad (6)$$

where $\Psi_\theta$ being the normalization constant. A sample $\mathbf{z}$ with high $q_\theta(\mathbf{z})$ from the latent chain will have low $H_\theta(\mathbf{z})$, and therefore its projection to the visible space $f_d(\mathbf{z})$ will have low $E(f_d(\mathbf{z}))$ by design.

Theoretically, running an infinitely long visible chain should suffice for generating samples from $p_\theta(\mathbf{x})$. As the choice of initial point has negligible effect on the convergence of the chain in theory, the introduction of the latent chain does not affect the asymptotic behavior of sampling.

Figure 5: Samples from AE (Left) and from EBAE (Right).

Figure 5 shows the samples generated from a conventional autoencoder and EBAE trained on MNIST through the proposed two-stage sampling method. The samples from the autoencoder are significantly different from MNIST digits. As samples with small reconstruction errors are more likely to be drawn, these non-MNIST images have smaller reconstruction errors than MNIST in-distribution images. Meanwhile, the samples from EBAE are visually similar to MNIST digits while also being diverse.

## 4.4 Technical Details in Training

Here, we describe a few technicalities for stable training and competitive performance with EBAE.

**Pre-training as a conventional autoencoder** Before training an EBAE using Eq.(3), we initialize the network parameters by training the network as a conventional autoencoder via reconstruction error minimization until convergence. Compared to computationally expensive LMC steps in EBAE training, the conventional autoencoder training is very stable and fast, also resulting in a good representation of data to start working with.

**Spherical latent space (Davidson et al., 2018; Xu & Durrett, 2018; Zhao et al., 2019)** The output of an encoder is projected to the surface of a unit ball through division by its norm. The spherical latent space removes the boundary effect for the latent chain in EBAE, improving the stability of LMC. To reflect the spherical constraint during LMC, we employ Constrained LMC Brubaker et al. (2012), where $\mathbf{z}_t$ is projected to the sphere after every LMC step (Eq.(4)).

**Regularization** We regularize the energy of negative samples to prevent its blow up. During the training, we minimize the average squared energy of negative samples in a mini-batch as well as the negative likelihood of data. The regularization term is given as $L_{reg} = \sum_{i=1}^{B} E(\mathbf{x}_i')^2/B$ for the batch size $B$, and its gradient $\alpha\nabla_\theta L_{reg}$ is added to the gradient of the negative log likelihood.

## 5 Related Work

**Autoencoder as a probabilistic model** While conventional autoencoders do not directly model a probability distribution, but some autoencoder variants have probabilistic interpretation. In DAE (Vincent et al., 2008) and contractive autoencoders (Rifai et al., 2011), the reconstruction error is related to the gradient of log density (Alain & Bengio, 2014). Generative stochastic networks (Bengio et al., 2014) utilize this property to build a model that can draw samples. Interestingly, Alain & Bengio (2014) reported that DAE might produce a spuriously small log density gradient estimate, i.e., a small reconstruction error, for a point in far from the support of data. VAE (Kingma & Welling, 2014) and its variants, including adversarial autoencoders Makhzani et al. (2015), WAE Tolstikhin et al. (2017), and Generative Probabilistic Novelty Detection Pidhorskyi et al. (2018), model a properly normalized probability density with the aid of a prior distribution.

**Outlier detection** Besides the reconstruction-based method mainly discussed in this paper (Japkowicz et al., 1995; An & Cho, 2015; Zong et al., 2018), autoencoders can be applied to outlier detection by learning representation of data. In Xu et al. (2015), input data are transformed using

Table 1: MNIST hold-out class detection results. AUC scores are shown. The values in parentheses denote the standard error of mean after 10 training runs.

| MNIST | 0 | 1 | 2 | 3 | 4 | 5 | 6 | 7 | 8 | 9 | avg |
|---|---|---|---|---|---|---|---|---|---|---|---|
| EBAE | **.989**$_{(.002)}$ | **.919**$_{(.013)}$ | **.992**$_{(.001)}$ | **.949**$_{(.004)}$ | **.949**$_{(.005)}$ | **.978**$_{(.003)}$ | **.938**$_{(.004)}$ | .885$_{(.024)}$ | **.929**$_{(.004)}$ | **.934**$_{(.005)}$ | **.946** |
| AE | .819 | .131 | .843 | .734 | .661 | .755 | .844 | .542 | .902 | .537 | .677 |
| DAE | .769 | .124 | .872 | .935 | .884 | .793 | .865 | .533 | .910 | .625 | .731 |
| VAE(R) | .954 | .391 | .978 | .910 | .860 | .939 | .916 | .774 | .946 | .721 | .839 |
| VAE(L) | .967 | .326 | .976 | .906 | .798 | .927 | .928 | .751 | .935 | .614 | .813 |
| WAE | .817 | .145 | .975 | **.950** | .751 | .942 | .853 | **.912** | .907 | .799 | .805 |
| GLOW | .803 | .014 | .624 | .625 | .364 | .561 | .583 | .326 | .721 | .426 | .505 |
| PXCNN++ | .757 | .030 | .663 | .663 | .483 | .642 | .596 | .307 | .810 | .497 | .545 |
| IGEBM | .926 | .401 | .642 | .644 | .664 | .752 | .851 | .572 | .747 | .522 | .672 |
| DAGMM | .386 | .304 | .407 | .435 | .444 | .429 | .446 | .349 | .609 | .420 | .423 |

the encoder in an autoencoder, then other outlier detection algorithms, such as one-class support vector machines (Schölkopf et al., 2001), are applied. Deep autoencoding Gaussian mixture model (DAGMM, Zong et al. (2018)) trains an autoencoder jointly with a mixture of Gaussian distributions defined in the latent space of the autoencoder. The key assumption of using the latent representation of an autoencoder is that an outlier will reside far from the training samples in the latent space.

After Nalisnick et al. (2019); Hendrycks et al. (2019) reported that deep generative models are not able to detect obvious outliers, a number of outlier detection methods have been proposed to remedy the problem. Some of those methods require training of multiple generative models (Ren et al., 2019; Choi et al., 2018). Other methods are highly specialized to image data (Golan & El-Yaniv, 2018; Tack et al., 2020; Serrà et al., 2020). In this paper, we focus on the generative approach which is more generally applicable.

**Energy-based models** The energy-based formulation (Eq.(1), Mnih & Hinton (2005); Hinton et al. (2006)) to model a density function has a long history of research. Recently, a number of works attempt to use a deep neural network to model an energy function, showing promising results on generative modeling and OOD detection. Du & Mordatch (2019); Grathwohl et al. (2020) use a deep feed-forward network as energy function and train it via contrastive divergence learning which involves LMC sampling. Zhao et al. (2016) propose Deep Structured Energy-Based Model (DSEBM), where the gradient of its energy function is set as the reconstruction error. Energy-based Generative Adversarial Networks (EBGAN) Zhao et al. (2016) employs the reconstruction error of a deep autoencoder as a discriminator and call it an energy function. However, EBGAN does not explicitly model a density function and trained via methods similar to other generative adversarial networks. Also, EBGAN requires a separate generator to generate samples.

# 6 EXPERIMENTS

## 6.1 EXPERIMENTAL SETUP

We empirically compare EBAE to other generative models in outlier detection tasks. In experiments, each method is trained using inlier data and then asked to discriminate outliers from inliers during test phase. As described in Section 2.1, we assume that a detection method produces a scalar decision function $c(\mathbf{x})$ given an input $\mathbf{x}$, and its performance is measured in AUC. Following the protocol of Ren et al. (2019) and Hendrycks et al. (2019), we use an additional OOD dataset different from the datasets used in test phase to tune model hyperpamraeters. Additional details on model implementation and datasets can be found in the supplementary material.

Our baselines includes autoencoders, deep generative models, and the state-of-the-art energy based model. We test conventional autoencoders (AE), DAE, VAE, and WAE as our autoencoder baselines. Additionally, DAGMM (Zong et al., 2018), an autoencoder-based approach which considers reconstruction error and density in the latent space simultaneously in outlier detection is tested. For all autoencoder-based methods including EBAE, we use the identical network architectures and vary other hyperparameters including the latent dimensionality. For AE, DAE, WAE, and EBAE, we use reconstruction error as $c(\mathbf{x})$. Since both reconstruction error and likelihood is available from VAE, we test both options for $c(\mathbf{x})$ and denote the results as VAE(R) and VAE(L), respectively. Deep

generative model baselines, PixelCNN++ (PXCNN++, Salimans et al. (2017)) and Glow (Kingma & Dhariwal, 2018) are trained by maximizing likelihood, and their negative log likelihoods are used as $c(\mathbf{x})$. We implement a deep energy-based model (IGEBM, Du & Mordatch (2019)) and use its energy as $c(\mathbf{x})$.

In addition to widely used datasets, including MNIST, FashionMNIST (FMNIST), CIFAR-10, SVHN, CelebA, and ImageNet resized to $32\times32$ (ImageNet32, Oord et al. (2016)), we additionally utilize two synthetic image datasets, Constant and Noise. In Constant dataset, all pixels of an image has the same value uniform-randomly drawn from the set $\{0, ..., 255\}$. Images in Constant dataset are gray, i.e., their channels have the same values. An image in Noise dataset has its pixel values independently drawn from the uniform distribution on the set $\{0, ..., 255\}$. Constant and Noise images are generated in two sizes, $1 \times 28 \times 28$ and $3 \times 32 \times 32$. Note that both image datasets are clearly OOD for all mentioned datasets. Pixel values of images are scaled to the $[0, 1]$ in all experiments.

## 6.2 HOLD-OUT CLASS DETECTION

We set one class from MNIST as the outlier class and the rest as the inlier class, and repeat the procedure for all classes in MNIST. This experiment simulates the situation where an inlier distribution consists of multiple disconnected clusters. Constant dataset of size $28\times28$ is used for select the best hyperparameters for each method. After hyperparameter selection, the training for EBAE under the selected hyperparameter is repeated for 10 different random seeds to demonstrate the variance of EBAE training. The results are shown in Table 1.

EBAE shows the highest AUC score for all classes except for 7, while still achieves the second best performance for detecting 7 and the best overall performance. Note that detecting 1, 7, or 9 as outlier is more difficult than detecting other digit. From the discussion in Section 3, we suspect that 1, 7, and 9 are more readily represented by composition of sub-patterns in other digits.

## 6.3 OOD DETECTION

In this experiment, we train generative models on a CIFAR-10 or ImageNet $32\times32$ (ImageNet32), and test how well they classify images from other datasets as OOD. Here, we use MNIST images padded with zeros to make 32x32 for model selection. Other details are covered in the supplementary material. Results are shown in Table 2 and Table 3. Outlier reconstruction is severe for Constant and SVHN datasets, as noted from AUC lower than 0.5. The right panel of Figure 1 compares the reconstruction error distribution of Constant and SVHN from AE and EBAE trained on CIFAR-10, showing that the outlier reconstruction is effectively suppressed in EBAE.

Table 2: OOD detection (in-distribution: CIFAR-10).

| OOD | Constant | FMNIST | SVHN | CelebA | Noise |
|---|---|---|---|---|---|
| EBAE | **.923** | **.819** | **.818** | **.789** | 1.0 |
| AE | .006 | .650 | .175 | .655 | 1.0 |
| DAE | .001 | .671 | .175 | .669 | 1.0 |
| VAE(R) | .002 | .700 | .191 | .662 | 1.0 |
| VAE(L) | .002 | .767 | .185 | .684 | 1.0 |
| WAE | .000 | .649 | .168 | .652 | 1.0 |
| GLOW | .384 | .222 | .260 | .419 | 1.0 |
| PXCNN++ | .000 | .013 | .074 | .639 | 1.0 |
| IGEBM | .192 | .216 | .371 | .477 | 1.0 |

Table 3: OOD detection (in-distribution: ImageNet32).

| OOD | Constant | FMNIST | SVHN | CelebA | Noise |
|---|---|---|---|---|---|
| EBAE | **.966** | **.994** | **.985** | **.949** | 1.0 |
| AE | .005 | .915 | .102 | .325 | 1.0 |
| DAE | .069 | .991 | .102 | .426 | 1.0 |
| VAE(R) | .030 | .936 | .132 | .501 | 1.0 |
| VAE(L) | .028 | .950 | .132 | .545 | 1.0 |
| WAE | .069 | .991 | .081 | .364 | 1.0 |
| GLOW | .413 | .856 | .169 | .479 | 1.0 |
| PXCNN++ | .000 | .004 | .027 | .238 | 1.0 |

## 7 CONCLUSION

In this paper, we have investigated the unexpected reconstruction of outliers in autoencoders. To fix the problem, we have proposed a novel probabilistic view on autoencoders, which leads to a new training scheme. In our experiments, an autoencoder trained by the proposed method is less prone to reconstruct an outlier and is highly effective in outlier detection.

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
