# OpenReview forum: "Suppressing Outlier Reconstruction in Autoencoders for Out-of-Distribution Detection"
_ICLR.cc/2021/Conference — Reject_

### Official Review · AnonReviewer2 · 2020-10-21
**A variant of EBM. Lack of experiments to convince why the proposed EBM is better than existing ones.**

**Rating:** 4
**Confidence:** 5

**Review:**

The author propose a variant of energy based model to define the energy via \|x-f(x)\| where f is a network.  The empirical studies are provided on MNIST, CIFAR and ImageNet32, which shows some advantage of the proposed algorithms.

1.  It is not convincing to me why the proposed model should be better than existing energy based model. After defining the energy as \|x-f(x)\|, the remaining details of the algorithm are routines by following the known results. It is the nature of all EBMs to assign high density to training data and lower down the energy of the sampled data. Based on this intuition, several algorithms have been proposed further, such as CD and PCD back to decades ago. Then the question is why use exp(-\|x-f(x)\|) as energy should be better than exp(-g(x))?  What's the implicit prior we leverage here? Whether it is especially useful for OOD or generally true for all density model? If so, then why? otherwise, it looks not convincing that it can significantly outperform the standard energy based model if the model capacity is the same, which I don't find either in the paper. The authors only briefly mention the architecture for ImageNet32 experiments.  The authors should shorten the descriptions of known results on EBMs and focus more on why the defined energy is better than existing  on OOD.

2.  The experiments of other baselines are potentially flawed.  The reported numbers are worse than they should be. For example, on CIFAR10 vs CelebA, GLOW can achieve 0.9 AUC, as reported here https://arxiv.org/pdf/2006.09273.pdf.

3. The considered experiments are relative easier.  There are some more challenging experiments could be considered to make the results stronger. For example, CIFAR10 vs CIFAR100. Or doing the same setting as the experiments in Table 1 on CIFAR.

4. Related works. There are other AE based works on OOD/Anomaly detection are not discussed, such as
* Rate-Distortion Optimization Guided Autoencoder for Isometric Embedding in Euclidean Latent Space

5. There are lots of recent works discussing about likelihood is not a good metric for OOD. Whether the proposed works contradicts those results? If yes, which design makes it better than all other LLD based model?

---

### Official Review · AnonReviewer3 · 2020-10-24
**A decent work on AE outlier detection**

**Rating:** 5
**Confidence:** 4

**Review:**

A generative autoencoder outlier detection called EBAE is proposed by introducing fake samples which produce small  reconstruction error, yet being outliers, during training to enhance the AE outlier detectability. The fake sample generation is done in 2 stages via MCMC method. Number of experiments are carried out on various datasets and the results are promising. The paper approaches an interesting topic that why AE  may fail to detect OOD samples. I have the following concerns/comments:

-	My first question is about fake sample generation, it is not clear to me why the generated samples should be considered as outliers? According to section 4.4 AE is initialized by training it in a conventional way, so it is expected that the generated samples from this model to be close to the training samples, how do we guarantee that the fake samples which already have small reconstruction error are necessarily OOD?
-	The convergence happens when P(x) and P_theta(x) are equal; in other words, the model would represent data distribution; thus the expectation is that the generated samples would look like the real ones more and more as the training goes forward? Is that being observed? What iteration results does Figure 5 depict?
-	How many fake samples are augmented in each iteration? According to the abovementioned point, would you reduce it as training goes forward?
-	In general, thinking of some simpler approaches to do the same thing:
o	As mentioned in the paper, the fact that AE may well reconstruct an outlier could be because of a good generalization, which results in inductive biases; then, overfitting could be also helpful to avoid that. How do you argue about that? and do you have any results showing the proposed method could outperform overfitting.
o	The proposed method utilizes data augmentation, and generating them via the sampling scheme presented in this paper is one way. Could we improve OOD detection by other augmentation like those artificial done in Figure 1?
-	There are number of GAN based anomaly detection methods (e.g. GANomaly, AnoGAN,..) to estimate the distribution of data, how do you compare your method with them?
-	Regarding Figure 1:
o	It would be better to see the left and right results for the same dataset (either MNIST or CIFAR10).
o	By looking at the right plots, it is difficult to see any improvement  using EBAE?
-	In Figure 3, due to scaling it is difficult to confirm “the outlier reconstruction occurs only when D_z is large”.
-	In equation (1), p_theta(x) is defined according to E_theta(x), but in equation (2) (and also in supplementary document) it is changed to E(x), without theta subscript. Is it a typo? How do you discriminate them?
-	In section 4.3 Figure 4 is referred as Figure 5 mistakenly.
-	In table 2 and 3 please mention the metric (what the numbers in the table represent) in the caption.

---

### Official Review · AnonReviewer4 · 2020-10-27

**Rating:** 5
**Confidence:** 4

**Review:**

### Summary
The paper describes a new method for detecting outliers with deep autoencoders by suppressing the reconstruction of out-of-distribution data. The article first investigates the reasons why standard autoencoders (AEs) reconstruct outlier datapoints fairly well, and are therefore problematic when used to detect anomalies via the reconstruction loss. The main novel contribution is the Energy-based Autoencoder (EBAE), a variant of an autoencoder in which the reconstruction loss is directly used as an energy function, and therefore outliers should have high energy. This is done with a new gradient formulation that enforces normalization of the probability distribution by sampling from the learned model via a variant of Langevin Monte-Carlo sampling. This second term is supposed to punish the reconstruction of outliers. Results are shown on a MNIST holdout task (leave one class out), and for OOD detection in CIFAR-10 and a downsampled version of ImageNet, showing that EBAE learns to reconstruct samples from different datasets, as well as constant and noise inputs, with significantly higher reconstruction error than competing methods.

### Evaluation
The field of anomaly detection with deep autoencoders is relevant for a large part of the ICLR community. The paper contributes a novel method that improves outlier detection on the tested benchmarks, so the paper has some significance for the field of anomaly detection. The paper is overall relatively well written, although there are some sections with quite a few grammar mistakes (e.g. Sec. 4.2) that should be corrected, and neither the figures nor their captions are very clear. The originality of the paper is quite limited, because it is to a large part based on known observations, and adds an incremental improvement over previous approaches tackling the same problem. The methods and the related work are described in sufficient detail. In the section on outlier reconstruction it is not clear which of the proposed reasons for outlier reconstruction by AEs are novel insights, and which are known facts from the literature. Only for the first argument (smoothness) a source is quoted. The experiments are quite extensive in comparing to different algorithms and with multiple outlier datasets, but the interpretation of these results is very short, almost non-existent.

Overall I think this paper is about at the acceptance threshold. It is a nice but not a groundbreaking new method, and although the results on the tested benchmarks look good, these are relatively easy benchmarks for outlier detection, and so the relevance for real-world problems is not quite clear. In particular it is not clear if the sampling mechanism can deal with difficult outlier detection problems where inliers and outliers are similar (e.g. outliers produced via corruptions). Additionally I am concerned about the computational complexity of the method compared to other approaches. I think the paper would greatly benefit from a more exhaustive interpretation of the results.

**Pros**:
1. The paper shows a new method for outlier detection that improves existing AE-based techniques.
2. There are a lot of experiments with different datasets and different competing algorithms for the three main tested datasets (MNIST, CIFAR-10, and ImageNet32).
3. The experimental results show an advantage on the tested datasets both for class holdout and outliers from other datasets.

**Cons**:
1. Energy-based interpretations of autoencoders are well-known, so this is a rather incremental addition to the existing literature.
2. The method adds computational complexity through the sampling process, and it is not clear how many samples are needed for the method to become effective. The supplementary material provides the number of steps to generate a sample, but I could not find the total number of samples generated (maybe I missed it). The authors say that the complexity is OK for the tested datasets, but I would assume for larger images and more complex datasets this could become a big issue, because there will not be an efficient sampling of outliers in such cases.
3. There is almost no interpretation of the results. Why e.g. do almost all other methods have great difficulty with the constant inputs, but no problems with noise? Why are methods such as PXCNN++ performing so poorly on most datasets?
4. The benchmarks are rather simple, because the tested outlier datasets are quite different from the training set. Tests e.g. on corrupted versions of the training data would be interesting to test the power of the method. Robust anomaly detection, i.e. detecting outliers under the assumption that a small fraction of training points are anomalies could also prove challenging, since such datapoints might be reproduced by the sampling process (see e.g. Beggel, L., Pfeiffer, M., & Bischl, B. (2019). Robust anomaly detection in images using adversarial autoencoders. ECML (pp. 206-222); Choi, H., Jang, E., & Alemi, A. A. (2018). Waic, but why? generative ensembles for robust anomaly detection. arXiv preprint arXiv:1810.01392.) It would be good to talk more in the paper about the limitations of the method.
5. In the tables on outlier detection I was missing the reconstruction error for inliers as a reference.
6. The figures are not properly described in the captions, and some of them e.g. Fig. 2 cannot be understood without reading the full text.


### Additional Comments
1. In Fig. 1 (left) and Fig. 5 (right) it appears that all reconstructions of EBAE have much higher brightness than the original and reconstructions of AE. Why is that?
2. In Fig. 1 (right) it is unclear why the reconstruction loss of EBAE is overall so much higher than for AE (the axes are shifted). The figure suggests a pretty high loss also for inliers, compared to AE, that's why I am also asking to provide the inlier values for Tables 2 and 3.
3. Please improve the caption of Fig. 2, this is not understandable.
4. In section 3 (towards the end) I don't understand the following sentence: "The latent representation is more distribution for large D_z...". Please correct.
5. In Section 4.2. I am not sure about the reasoning why the sampling should produce outliers at all. It is only mentioned that sampling produces high-likelihood samples, but if only inliers are sampled, how does this help distinguishing from real outliers? It seems unlikely that proper outliers are sampled at all from this process.
6. Section 6.1, Typo: "hyperpamraeters"

---

### Official Review · AnonReviewer1 · 2020-10-28
**Paper is decently written but the experimental results section needs significant improvement**

**Rating:** 4
**Confidence:** 4

**Review:**

The authors address an important problem of autoencoders having low reconstruction error for OOD instances. They use the regular inlier reconstruction loss minimization with an additional term to maximize reconstruction loss for fake sampled OOD instances. A two stage Langevin Monte Carlo sampling technique is used for sampling in the proposed EBAE framework to generate diverse samples. Empirical analysis reveals that the proposed method outperforms existing autoencoder baselines. Some of my concerns with this paper are the following

a) This contribution "We present a sampling method tailored for EBAE which efficiently generates diverse samples" needs better clarification and presentation. Figure 5 conveys the message but the writeup and math need to be written more clearly.

b) Table 1 is not rightly placed. It should not be in related work section. Also why are standard deviation values not reported for baselines ? Are those values taken from the corresponding papers directly ? For anomaly detection in particular it is not sufficient just to report AUC and other metrics like AUPR or F-1 score should also be reported.

c) Table 2 and Table 3 the captions are not descriptive enough.

d) Another relevant autoencoder baseline which should be considered for empirical analysis is Feature Bagging Autoencoder. (Outlier Detection with Autoencoder ensembles. Chen et al. 2017)

---

### Public Comment · ~Jianwen_Xie1 · 2020-11-14
**about related works**

Dear Authors and Reviewers,

We found that the current paper missed some important references about pioneering works that are related to energy-based generative models parameterized with deep net energy.

The first paper that proposes to train an energy-based model parameterized by modern deep neural network and learned it by Langevin based MLE is in (Xie. ICML 2016) [1]. The model is called generative ConvNet, because it can be derived from the discriminative ConvNet. This is also the first paper to formulate modern ConvNet-parametrized EBM as exponential tilting of a reference distribution, and connect it to discriminative ConvNet classifier. That is, EBM is a generative version of a discriminator. (Xie. ICML 2016) [1] originally studied such an EBM model on image generation theoretically and practically in 2016.

(Xie. CVPR 2017) [2] (Xie. PAMI 2019) [3] proposed to use Spatial-Temporal ConvNet as the energy function in EBMs for video generation. The model is called Spatial-Temporal generative ConvNet.

(Xie. CVPR 2018) [4] also proposed to use volumetric 3D ConvNet as the energy function for 3D shape pattern generation. It is called 3D descriptor Net.

Also, the Generative Cooperative Nets (CoopNets) (Xie. PAMI 2018)[5] and (Xie. AAAI 2018) [6], which jointly trains an EBM and a generator network by MCMC teaching.

Those are the more original and earlier papers for deep EBMs with ConvNet as energy function than what you have cited, e.g., [7](Yilun Du and Igor Mordatch, 2019).

References:

[1] A Theory of Generative ConvNet. Jianwen Xie *, Yang Lu *, Song-Chun Zhu, Ying Nian Wu (ICML 2016)

[2] Synthesizing Dynamic Pattern by Spatial-Temporal Generative ConvNet Jianwen Xie, Song-Chun Zhu, Ying Nian Wu (CVPR 2017)

[3] Learning Energy-based Spatial-Temporal Generative ConvNet for Dynamic Patterns Jianwen Xie, Song-Chun Zhu, Ying Nian Wu IEEE Transactions on Pattern Analysis and Machine Intelligence (TPAMI) 2019

[4] Learning Descriptor Networks for 3D Shape Synthesis and Analysis Jianwen Xie *, Zilong Zheng *, Ruiqi Gao, Wenguan Wang, Song-Chun Zhu, Ying Nian Wu (CVPR) 2018

[5] Cooperative Training of Descriptor and Generator Networks. Jianwen Xie, Yang Lu, Ruiqi Gao, Song-Chun Zhu, Ying Nian Wu. IEEE Transactions on Pattern Analysis and Machine Intelligence (TPAMI) 2018

[6] Cooperative Learning of Energy-Based Model and Latent Variable Model via MCMC Teaching. Jianwen Xie, Yang Lu, Ruiqi Gao, Ying Nian Wu. AAAI 2018.

[7] Yilun Du and Igor Mordatch. Implicit generation and modeling with energy based models. In Advances in Neural Information Processing Systems, pages 3603–3613, 2019

Thank you!

---

### Decision · Program_Chairs · 2021-01-07
**Final Decision**

**Decision:**

Reject

**Comment:**

The paper proposes to use reconstruction error of autoencoder as the energy function and normalize the resulting density for detecting anomalous/OOD examples. Reviewers have raised several concerns with the paper, including, lack of insights into why the AE energy is better for OOD detection than other energy function parameterizations, and incremental nature of the proposed method. Authors have not responded to these concerns. The paper is not suitable for publication in its current form.